# Targeting Perineural Invasion in Pancreatic Cancer

**DOI:** 10.3390/cancers16244260

**Published:** 2024-12-21

**Authors:** Ingrid Garajová, Elisa Giovannetti

**Affiliations:** 1Medical Oncology Unit, University Hospital of Parma, 43126 Parma, Italy; 2Department of Medical Oncology, Lab of Medical Oncology, Cancer Center Amsterdam, Amsterdam UMC, VU University Medical Center (VUmc), 1007 MB Amsterdam, The Netherlands; e.giovannetti@amsterdamumc.nl; 3Cancer Pharmacology Lab, AIRC Start-Up Unit, Fondazione Pisana per la Scienza, San Giuliano Terme PI, 56017 Pisa, Italy

**Keywords:** pancreatic ductal adenocarcinoma, neural invasion, tumor microenvironment, cancer pain

## Abstract

Pancreatic cancer is a leading cause of cancer death worldwide with an increasing incidence. Neural invasion is present in almost all pancreatic cancers. Clinically, it is translated into intractable pain and worse outcomes. Here, we highlight the importance of peripheral nerves in the pancreatic tumor microenvironment influencing the initiation and dissemination of pancreatic cancer. Furthermore, both autonomic (sympathetic and parasympathetic) and afferent nerves modulate different signaling pathways promoting tumor survival and immune escape. For this reason, exploring the potential synergistic benefits of anti-neurogenic therapies combined with chemotherapy or immunotherapy might inhibit both pancreatic cancer progression and alleviate cancer-related pain.

## 1. Introduction

In humans, the nervous system consists of two main components: the peripheral nervous system (PNS) and the central nervous system (CNS). Generally, nerves are efferent (motor) when they transmit signals from the brain or afferent (sensory) when they transmit information from the body to the CNS. The PNS consists of peripheral nerves that connect any part of the human body with the CNS which consists of the brain (the starting place of cranial nerves, including the nervus vagus) and spinal cord (the starting place of spinal nerves). The PNS is further divided into somatic and autonomic nervous systems. The somatic nervous system consists of nerves that link the CNS to skeletal muscles and to skin’s sensory receptors. The autonomic nervous system is responsible for internal organs’ innervation and is further subdivided into sympathetic, parasympathetic, and enteric nervous systems. The nerve fibers of the sympathetic nervous system are derived from the lateral horn of the spinal cord whilst parasympathetic nervous system fibers originate from the brainstem [1,2]. Generally, the parasympathetic nervous system is dominant when organisms are in a relaxed state; on the other side, the sympathetic nervous system is activated in critical or stressed situations in order to mobilize energy to fight. In addition, the enteric nervous system aims to control the gastrointestinal system [3,4,5,6]. At a cellular level, nervous tissue consists of neurons. Almost all neurons have a body (with a nucleus), dendrites, and axons [3,7,8]. Apart from neurons, an important component of the nervous system is glial cells that give to neurons structural and metabolic support [3,5,7,8]. In the PNS, glial cells include Schwann cells (SCs) and satellite cells, and in the CNS, they include astrocytes, oligodendrocytes, ependymal cells, and microglia. In pancreatic ductal adenocarcinoma (PDAC), nerve fibers include axons originating both from autonomic nerve fibers (parasympathetic and sympathetic) as well as from afferent (sensory vagal and spinal) nerve fibers and from the enteric nervous system (Figure 1). Moreover, newly developed nerve fibers are present in the PDAC microenvironment [9]. Sympathetic nerve fibers mainly influence blood vessels and islets. Parasympathetic fibers innervate islets and intrapancreatic ganglia. Finally, afferent sensory neurons innervate the exocrive pancreas and islets. Intrapancreatic ganglionic neurons receive inputs from all types of nerve fibers [10,11].

## 2. Clinical Significance of Neural Invasion in Pancreatic Cancer

Solid tumors disseminate in three ways: the lymphatic, haematogenic, or direct invasion of surrounding tissue. As explained in the previous chapter, the PNS can be viewed as a neuronal circuit that connects all body parts and organs to the CNS in order to regulate muscle movements and sensations [3,12]. Nerves also play a trophic function for the epithelium; they are involved in the development of epithelial organs as well as their functional maintenance [13]. This role is fundamental both in healthy and cancer tissues. More nerves are needed for more cancer cells. More recently, their role in pancreatic tumorigenesis has been elucidated and presents a fourth route of cancer dissemination [14,15,16]. Cancer cells control nerves by the induction of axonogenesis (the enlargement of nerves), neurogenesis (the growth of neural progenitors), and neural reprogramming (the transformation of a sensory nerve into an adrenergic nerve) [17]. In this way, nerves can provide nutrition for new malignant epithelial cells. In fact, neural invasion (NI) is present in several solid tumors with the highest prevalence in PDAC ranging between 70% and 98% [18]. NI is defined as the presence of tumor cells in different spaces of the neuron sheath (endoneurium, perineurium, epineurium) or along nerves if tumor cells circumscribe at least 33% of the nerves [15,19,20]; see Figure 2. From a clinical point of view, NI is translated into intractable pain and represents a predictor of tumor recurrence and poor prognosis [21,22,23].

It is estimated that approximately 80% of patients with PDAC report moderate to severe pain intensity levels [24]. Different molecules are involved both in the process of NI and pain onset, for example, nerve growth factor (NGF) which belongs to the family of neutrophins [25,26,27,28]. NGF is a factor essential for the nutrition and survival of sympathetic and sensory neurons in vivo and in vitro. NGF produced by tumor cells can bind to afferent nerves, especially to transient receptor potential vanilloid 1 (TRPV1) [29,30,31]. TRPV1 is an ion channel which modifies the permeability of several ions (Ca^2+^, H^+^, Na^+^, Mg^2+^) [29,30,31]. The activation of this ion channel is important for neurotransmitters’ release from sensory nerves (for example, substance P (SP) and calcitonin gene-related peptide (CGRP)) which, in consequence, leads to severe pain [29,30,32]. NGF acts on two other receptors with different affinities: p75NTR and TrkA. As demonstrated in Figure 3 and Figure 4, NGF binding to p75NTR (low-affinity receptor) leads to the activation of the nuclear factor-κB (NF-κB) and c-Jun N-terminal kinase (JNK) signaling pathways. NGF binding to TrkA (high-affinity receptor) activates several signaling pathways: PLCγ, PI3K/AKT, and RAS/MAPK. Several more molecules belonging to groups of neutrophines play an important role in nerve growth and survival, in particular brain-derived neurotrophic factor (BDNF), neurotrophin-3 (NT-3), and neurotrophin-4,5 (NT-4/5). The TrkB receptor binds to both BDNF and NT-4, TrkC binds to NT-3, and the GFR-ɑ1-RET receptor binds to GDNF [28,33]. NGF and TrkA are expressed in both PDAC cells and nerves, which strongly suggests their involvement in NI. Importantly, NGF and its high-affinity receptor TrkA have been found to be overexpressed in PDAC tissues compared to in a normal pancreas [23]. However, no difference in NGF and TrkA expression was observed between early and more advanced PDAC stages nor between lower and higher tumor grades. Nonetheless, tumors with NGF/TrkA overexpression exhibited an increased incidence of NI and were associated with a higher degree of pain [23]. This correlation was further confirmed by Dang et al., who demonstrated that TrkA overexpression is linked to NI, more severe pain, and poorer outcomes [34]. In addition, TrkA has been associated with NI in human tissue compared to tissue without NI. Several authors have demonstrated that NGF released by tumor cells promotes neuritic growth [35,36,37,38], reduces apoptosis, and increases the proliferation of cancer cells, leading to PDAC aggressiveness [37,39,40,41,42]. Additional studies have reinforced the notion that the overexpression of NGF and TrkA correlate with a higher incidence of NI, increased pain, and worse survival outcomes in PDAC patients [26,28,38,41,42,43,44]. Moreover, NGF/TrkA overexpression has been correlated with lymph node metastases [45]. Tissue TrkA mRNA expression levels have been higher among PDAC patients with more NI, more severe pain, and poorer outcomes [25,34,46]. In contrast, high p75NTR mRNA expression has been associated with a more favorable outcome [25,34,46]. Interestingly, hyperglycaemia may precede the diagnosis of PDAC by an average of 36–30 months [47]. In PDAC, a tumor microenvironment with elevated glucose levels can promote the overexpression of NGF and its receptor, thereby synergistically promoting the occurrence of NI [48,49]. In fact, increased NI has been found to correlate with hyperglycemia in PDAC patients [48,50].

## 3. The Role of Nerves in the Pancreatic Tumor Microenvironment

The tumor microenvironment (TME) is crucial for cancer progression. In PDAC, cancer cells interact with nerves and regulate nerve growth. This growth induces the upregulation of nerve survival-related protein expression, axonogenesis, neurogenesis, and neural reprogramming, increasing nerve density and the formation of new nerves in the tumor. The new nerves can induce the immune escape and survival advantage of the tumor cells [51]. According to Amit et al. [52], NI goes through seven steps: cancer cell survival, nerve homeostasis, inflammatory response, cancer cell chemotaxis towards the nerves, neurogenesis, cancer cell adhesion to nerve sheaths, and nerve invasion. The TME is composed of different cell types that are involved in both tumorigenesis and NI, thus supporting the existence of an intimate nerve–cancer interaction (Figure 5).

Most studies concur that PDAC is characterized by a predominantly immunosuppressive tumor microenvironment (TME), primarily driven by tumor-associated macrophages (TAMs) and myeloid-derived immunosuppressive cells (MDSCs). Intriguingly, the NGF pathway plays a role in TME remodeling mediated by TAMs and MDSCs in PDAC [53,54]. Understanding the precise changes within the TME and targeting the associated molecular pathways could play a crucial role in developing new treatments for PDAC.

### 3.1. Schwann Cells

The Schwann cell (SC) is the principal type of glial cell in the PNS. SCs are essential components within the tumor microenvironment; they provide nutritional support for neurons and initiate regeneration processes after injury [13,52]. The regeneration process after injury is identical to what can be seen in cancer patients; it initiates with the activation of some SCs. They become “repair SCs” (i.e., nonmyelinating, GFAP+) [55]. This was observed by Deborde et al. who reported more repair SCs close to pancreatic cancer cells [56]. This proximity to cancer cells leads to direct interactions with SCs via adhesion molecules such as NCAMs. After close contact, SCs can intercalate between tumor cells, leading to tumor cell dissemination along peripheral nerves [56]. SCs secrete chemokines (e.g., CCL-2). These substances are attractive for immune cells, in particular for macrophages. Macrophages product collagenases. They ultimately worsen neural injury by perineurium degradation and increase cancer cell penetration via the damaged perineurium. Therefore, neural damage becomes a loop of continuous damage processes performed by cancer and inflammatory cells [57]. The result of these interactions is NI promotion in vitro and in vivo [52,56,58]. Additionally, indirect interaction between SCs and neurons exist though secretory proteins such as L1CAM and transforming growth factor-beta (TGF-β).

### 3.2. Macrophages

The interactions among cancer cells, nerves, and immune cells promote cancer escape. Not only SCs attract macrophages in the site of NI, but cancer cells also secrete colony-stimulating factor 1 (CSF- 1) to recruit macrophages. Tumor-associated macrophages (TAMs), a pro-tumor M2 subtype of macrophage, release GDNF, contributing to NI promotion [54]. Furthermore, TAMs release leukemia inhibitory factor (LIF), which promotes SC migration and neural plasticity [59]. Not surprisingly, immunohistochemical staining showed the association of the presence of TAMs and NI in PDAC patients and an increased number of TAMs was correlated with the poor survival of PDAC patients [60]. Moreover, TAMs can upregulate metalloproteinases (MMPs) which cause damage to the extracellular matrix around peripheral nerves [61,62]. The overexpression of MMP9 is associated with worse clinical outcomes for PDAC patients [61,62].

### 3.3. Fibroblasts and Pancreatic Stellate Cells

In the PDAC TME, cancer cells have different neighbors, including pancreatic stellate cells (PSCs) and fibroblasts. An important protective barrier of peripheral nerves, perineurium, is composed of fibroblasts. Changes in the TME also affect fibroblasts that in consequence might lead to the modification of perineurium permeability and its increased vulnerability which is connected with the increased incidence of NI [49]. Cancer-associated fibroblasts (CAFs) produce several cytokines and growth and pro-angiogenic factors and contribute to angiogenesis and tumor progression [63,64]. Interestingly, CAFs themselves stimulate neural remodeling and SCs’ proliferation by the secretion of different molecules, such as Slit2 [65]. The principal fibroblasts in the pancreatic mesenchyme are PSCs. PSCs are activated during tumorigenesis and exploit a pro-tumorigenic influence on tumor dissemination [66,67]. These cells play an important role in neural growth and tumor cell migration in proximity to nerves.

### 3.4. Endothelial Cells

PDAC is known to be a hypoxic tumor. Hypoxia induces the secretion of hypoxia inducible factor-1α (HIF- 1α). HIF- 1α forces vascular endothelial growth factor (VEGF) production through the induction of different chemokines (CXCL12, CXCR4, and CX3CR1), resulting in new blood vessel formation, a process called neo-angiogenesis [12,68,69]. Complex interactions between cells within the tumor microenvironment facilitate cancer cell survival. Peripheral nerves in the TME release neurotransmitters (catecholamines, acetylcholine, SP), and cancer cells release several growth factors. However, this bidirectional interaction modifies the entire tumor microenvironment as released growth factors and neurotransmitters act on immune and endothelial cells, leading to chronic inflammation and neo-angiogenesis to provide nutritional support to new malignant cells [12,68,69].

## 4. Molecules Involved in Neural Invasion

The reciprocal interaction between cancer cells and the surrounding TME play an important role in tumor pathogenesis. It is initiated and driven by numerous molecular signals that promote cancer progression including neurogenesis and axonogenesis, contributing to the complex interplay between cancer cells and the neural components of the TME. Several molecules are involved in neural invasion in PDAC, including neutrophins, adhesion factors, proteins, chemokines, axon-guidance molecules, and, finally, neurotransmitters and neuropeptides (Figure 6).

### 4.1. Neutrophins

Neurotrophins are a group of proteins that promote the growth, function, and survival of neurons [12]. This group consist of several molecules: NGF, brain-derived neurotropic factor (BDNF), glial cell line-derived neurotrophic factor (GDNF), neurotrophin-3 (NT-3), and neurotrophin-4/5 (NT-4/5) [70]. NGF is an essential trophic molecule for sympathetic and sensory neurons in vivo and in vitro. The denervation of peripheral organs lead to a decrease in NGF production and is responsible for sensory and sympathetic ganglia hypoplasia [25]. Importantly, not only peripheral tissues but also cancer and neuronal tissues are rich sources of NGF [16]. NGF binds to p75NTR and TrkA receptors. Additionally, in afferent nerve fibers, NGF may stimulate transient receptor potential cation channels subfamily V member 1 (TRPV1) [30,33]. The upregulation of the TrkA receptor together with NGF is associated with increased NI intensity, poor outcomes, and more severe pain in PDAC patients [26,44]. BDNF and NT-3, with their receptors, TrkB and TrkC, respectively, have been found to be expressed in different solid tumors and are again associated with cancer progression and dissemination. Both BDNF and NT-3 are secreted by tumor and neuronal cells [16]. Lastly, the GDNF family consists of four proteins: GDNF, neurturin (NTN), artemin (ART), and persephin (PSP). They bind to four different types of GDNF receptor-α (GFRα) that signal through the tyrosine kinase receptor RET. GDNF is released by macrophages, SCs, and motor neurons. The upregulation of GDNF correlates with NI in PDAC [57,71]. GFR-ɑ2 is a receptor for NTN and has a role in the development of parasympathetic nerve fibers in PDAC. The overexpression of GFR-ɑ2 receptor and NTN is associated with neural plasticity and cancer progression [72,73]. Additionally, the GFR-ɑ3 receptor and ARTN have been shown to be overexpressed in PDAC and correlated with increased NI. PSP is released by macrophages, SCs, and motor neurons and is involved in NI and metastasis promotion [16,74].

### 4.2. Adhesion Factors

The association of neural cell adhesion molecules (NCAMs, also known as CD56) with the intensity of NI in PDAC has been demonstrated. In particular, NCAMs are an important mediator between SCs and cancer cells [75,76]. The process when repaired SCs promote NI by disseminating cancer cells into neurons is dependent on NCAM expression in SCs. As expected, NCAM-deficient mice showed less NI with less aggressive tumors [56]. Another group of adhesion molecules playing a role in NI are those in the claudin (CLDN) family. They are responsible for junctions between cells and are recognized as overexpressed in different tumor types including PDAC. Indeed, changing the cell junctions between cells might lead to the loss of cell–cell adhesion, which promotes tumor dissemination [45]. Their overexpression has been correlated with aggressivity and increased NI in head and neck tumors [77]. For their recognized role in cancer pathogenesis, they have been extensively studied as potential therapeutic targets (in particular, claudin-18.2, claudin-4, and claudin-7), including in PDAC [78].

### 4.3. Proteins

These include the vast family of matrix metalloproteinases (MMPs). They are a group of endopeptidases that have the capacity to degrade almost every component of the ECM and therefore to provoke ECM remodeling. The MMP family has upregulated expression in PDAC and it is associated with the occurrence of NI, tumor progression, and poor prognosis [79,80,81].

### 4.4. Chemokines

Chemokines are a group of peptides secreted by several non-cancer cell types such as immune cells and endothelial and epithelial cells, as well as by tumor cells [80,81,82]. They are best known for their ability to stimulate the migration of cells. They are divided into four subgroups (C, CC, CXC, and CX3C) [83]. Chemokines unleash their broad biological functions through cell surface G protein-coupled chemokine receptors (GPCRs). For each subgroup of chemokine, different receptors exist: C receptor (CR1), CC receptor (CCR1-10), CXC receptor (CXCR1-7), and CX3C receptor (CX3CR). Chemokines orchestrate inflammatory and immune responses and are responsible for attracting inflammatory cells in the proximity of injury. In PDAC, several of these chemokine ligands/receptors are correlated with NI, particularly CX3CL1/CX3CR1, CCL2/CCR2, CXCR4/CXCL12, CCL5/CCR5, and CXCL13/CXCR5 and signaling axes [84,85,86]. 

### 4.5. Axon-Guidance Factors

Axon-guidance factors include several families such as Slits, Semaphorins, Netrins, and Ephrins. They can be found in the tumor microenvironment. They exhibit a dual role, both of the inhibition and promotion of nerve guidance. In clinical oncology, Semaphorins and Slits are of particular interest. Semaphorins (such as SEMA3D and SEMA4D) are characterized by aberrant expression in several cancer types and contribute to cancer initiation. In PDAC, the overexpression of SEMA3D has been demonstrated. Moreover, EMA3D promotes NI and cancer dissemination in PDAC cells [65,87,88,89]. Slits (SLIT1, SLIT2, and SLIT3) are a protein family that act on ROBO receptors (ROBO1, ROBO2, ROBO3, and ROBO4) [90,91]. Together, they are involved in cell interactions in their tumor microenvironment [90,91]. In PDAC, both Slit glycoproteins and their ROBO receptors are associated with NI and cancer progression in PDAC [92,93,94]. Slit glycoprotein-2 inhibits tumor metastasis and NI in PDAC [87].

### 4.6. Neurotrasmitters and Neuropeptides

Peripheral nerves release neurotransmitters which act on cancer cells in a paracrine manner. However, the neurotransmitters enter the tumor microenvironment and influence a variety of different cells with corresponding receptors and act as tumor progression regulators [49]. As previously described, TRPV1 receptors are expressed in sensory neurons. Interestingly, they might also be present in non-neuronal tissues. TRPV1 activation leads to ion channel opening with increasing permeability for ions, including calcium. This process leads to the release of neuropeptides (e.g., SP, CGRP). SP increases vascular permeability and provokes vasodilation and edema. Moreover, it can activate mast cells with the subsequent release of a cascade of pro-inflammatory molecules such as histamine. Apart from mast cells, the activation of leucocytes to produce proteases and reactive oxygen species (ROS) potentiate inflammatory reactions. CGRP relaxes arteries and leads to inflammatory processes. Both neuropeptides, SP and CGRP, play an important role in pain [16,95]. Indeed, TRPV1 overexpression is found in PDAC patients suffering pain [29]. SP directly promotes tumor progression and NI in PDAC [96]. Importantly, both SP and its receptor neurokinin-1 (NK-1R) are overexpressed in PDAC tissues. SP promotes tumor proliferation, dissemination, and an increase in NI through the activation of the NK-1R/Akt/NF-κB signaling pathway [97].

## 5. Potential Therapeutic Targets in Perineural Invasion

To date, anti-neurogenic therapies have primarily been explored for managing pain and neurological diseases; however, these findings cannot be directly applied to oncology for inhibiting nerve infiltration in cancer. Although some data exist in the oncology context, most strategies targeting neural invasion have focused on the NGF-TrkA signaling pathway.

### 5.1. Targeting NGF-TrkA Signaling Pathway and Another Neutrophins

Neural invasion impacts quality of life in PDAC patients and therefore presents a promising therapeutic target. Prenatally, the absence of NGF leads to insensitivity to pain due to the absence of NGF-mediated nerve growth. Several genetic causes have been diagnosed in TrkA or NGF genes. TrkA gene and NGF gene mutation causes congenital insensitivity to pain together with anhidrosis due to the lack of sympathetic and nociceptive nerves [33]. Indeed, the activation of the NGF/TrkA signaling pathway is responsible for pain, and therefore, either NGF or TrkA is a potential therapeutic target for alleviating cancer-associated pain [98,99]. This pathway can be inhibited at different levels: (1) inhibiting NGF, (2) inhibiting NGF binding to TrkA, (3) TrkA inhibitors, and (4) inhibitors of downstream signaling pathways (Figure 7). This approach has been investigated in several preclinical in vivo models. Jimenez-Andrade et al. injected prostate cancer cells into mouse bone. As a consequence, they observed the pathological sprouting of sensory, CGRP-containing nerve fibers into the vicinity of cancer cells [100]. Almost all sensory nerves that underwent sprouting showed TrkA-positivity. The authors administered an NGF-blocking antibody which led to pain relief [100,101]. Similar experiments were conducted by Bloom et al. The authors injected human breast cancer cells into mouse bone. As in the previous study, the sprouting of sensory nerves was observed. The sprouting nerves were all CGRP-positive, underlining the role of this neuropeptide in the pathogenesis of cancer-associated pain. Moreover, the expression of TrkA and the growth-associated protein-43 (GAP43) was demonstrated. Therapeutic intervention with anti-NGF antibody administration had two effects: the blockage of nerve sprouting and pain relief [102]. The authors concluded that blocking the NGF/TrkA axis might have had a positive effect on pain relief due to the blockage of sensory nerve sprouting. The third type of tumor to prove the importance of NGF/TrkA blockage as a possible target for pain relief is sarcoma. In a study, larotrectinib (an inhibitor of TrkA, B, and C), improved cancer-associated pain and sensory nerve remodeling. Moreover, according to the study, the timing of TrkA blockage was important; in particular, the earlier the blockage occurred, the more effective the relief of cancer-associated pain and the tumor-induced remodeling of sensory nerve fibers were [103]. Several clinical studies on the efficacy of NGF antibodies have been conducted in a rheumatologic setting, in particular with tanezumab, fasinumab, and fulranumab with painful conditions such osteoarthritis and chronic low back pain [104,105,106,107,108,109,110,111,112,113,114,115,116] (Table 1). After the completion of a phase III study with tanezumab, this drug did not proceed in oncology settings due to collateral effects (osteonecrosis and adverse effects on the autonomic nervous system) [117].

### 5.2. Targeting Adhesion Factors and Proteins

It has been found that L1CAM secreted from SCs acts as a strong chemoattractant to cancer cells, through the activation of MAP kinase signaling. Additionally, L1CAM upregulates the expression of MMP-2 and MMP-9 by PDAC cells, through STAT3 activation. In in vivo mouse models, treatment with an anti-L1CAM antibody significantly reduces NI in vivo [119]. The MMP family includes matrix metalloproteinases (MMPs), a disintegrin and metalloproteases (ADAMs), and a disintegrin and metalloproteinase with thrombospondin motifs (ADAMTSs); they are all involved in ECM remodeling and degradation activities and might be blocked by different inhibitors [120,121,122]. Xu et al. investigated the function and underlying mechanisms of MMP1 in NI in PDAC [123]. They found that the silencing of MMP1 prevented PDAC cells from EMT and Schwann-like cell differentiation via inhibiting the activation of the NT-3/TrkC signaling pathway, thus alleviating NI in PDAC. The CLDN family plays an important role in cell junctions and has been found to be increased in several tumor types, including pancreatic cancer. As expected, the low expression of CLDN4 mRNA was correlated with more aggressive disease due to accelerated epithelial cancer cell dissemination. On the other hand, CLDN4 mRNA overexpression was associated with better survival in PDAC patients [78,124]. Interestingly, reduced CLDN4 expression was associated with increased NI in the study [124].

### 5.3. Targeting Chemokines

Several chemokines have been studied in tumors. For pancreatic cancer, some axes seem to have particular importance. The CXCL12/CXCR4 axis is the most important member of the chemokine family. The overexpression of CXCR4 has been found in PDAC cells in mouse models. In the same study, DRG overexpressed CXCL12 [125]. In another study, the above-mentioned axis correlated with increased MMP-2 and MMP-9 expression, corresponding to more aggressive disease [125]. Demir et al. identified CCL21/CXCL10 as key mediators of neural remodeling, NI, and pain in pancreatic cancer [126] and therefore presented them a potential therapeutic target. Indeed, the CXCR4-specific inhibitor AMD3100 has been evaluated by different research groups. In both in vitro and in vivo analyses, inhibiting CXCR4 resulted in the inhibited metastatic potential of cancer cells [125]. Moreover, anti-CCL2 antibodies have been investigated in preclinical colorectal cancer models. The inhibition of this chemokine leads to the inhibition of NI as well as the inhibition of angiogenesis through Akt and MAPK signaling pathways [127]. Another indicator of the severity of NI in PDAC is the overexpression of CX3CL1 and CX3CR1 [128]. Thus, this axis might also become a valuable therapeutic target contrasting neural dissemination in PDAC.

### 5.4. Targeting Axon-Guidance Factors

The axon-guidance factors most studied in PDAC are SEMAs, together with Slit glycoproteins. In particular, SEMA3D, with its plexin D1 (PLXND1), has been linked with the invasion and dissemination of pancreatic cancer cells. Moreover, its correlation with NI and overexpression in human PDAC tissues underline the importance of axon-guidance molecules [89]. This target has been evaluated in early phase 1 studies in solid tumors evaluating the safety of an anti-SEMA4D antibody (Pepinemab) [129,130]. Finally, Slit proteins and their ROBO have been demonstrated to have a role in cancer neural invasion and cancer progression in PDAC [87,91,94]. Slit glycoprotein-2 inhibits tumor metastasis and NI in PDAC [65,87].

### 5.5. Targeting Neurotrasmitters and Neuropeptides

Several drugs modulating the autonomous nervous system already exist. Regarding the sympathetic nervous system, a group of selective or non-selective β-blockers (propranolol or metoprolol) modulate sympathetic signals. The combination of β-blockers with gemcitabine has indeed reduced NGF expression and nerve density and improved the survival rate of KPC mice [131]. Regarding the modulation of the parasympathetic nervous system, blocking cholinergic receptor muscarinic 3 (CHRM3) by darifenacin inhibited prostate cancer growth in vitro and in vivo [132]. It seems that darifenacin might confer resistance to chemotherapy in PDAC. Indeed, CHMR3 overexpression has been associated with resistance to gemcitabine [133,134]. Several authors describe parasympathetic-like drugs as having a cancer-associated pain-relief effect [135,136]. Some data confirm that resiniferatoxin (TRPV1 agonist) could promote the apoptosis of pancreatic cancer cells and induce pain relief [29,137,138]. In recent years, two classes of drugs antagonizing CGRP have therefore been developed as the first migraine-specific preventive treatments: anti-CGRP monoclonal antibodies (erenumab, galcanezumab, fremanezumab, and eptinezumab) and gepants (rimegepant and atogepant). These new drugs have demonstrated efficacy and safety in clinical trials for both episodic and chronic migraines [139,140,141,142,143,144,145]. Currently, there are many data demonstrating the involvement of the SP/NK-1R system in cancer; therefore, NK-1R antagonists might represent a new antitumor strategy. As an example, an NK-1R antagonist, aprepitant, is already used in oncology as an antiemetic [146]. A correlation between NK-1R upregulation in cancer cells and the activation of the SP/NK-1R system with the progression of multiple cancer types and poor clinical outcomes has been shown by different studies [147,148]. Beirith et al. demonstrated that the inhibition of NK-1R by aprepitant achieves a significant reduction in the cell growth of pancreatic cancer cells, cancer stem cells, and pancreatic stellate cells in a dose-dependent manner. Interestingly, cancer cells expressing higher levels of the truncated tachykinin receptor 1 (TACR1) isoform exhibit more important sensitivity to NK-1R inhibition which might enable the stratification of PDAC patients who could benefit from NK-1R-targeted therapies [149]. Considering the impact of nerves on other components of the TME, in particular on immune cells (e.g., T cells, macrophages, NK cells, MDSC cells), it might be interesting to explore the potential synergistic benefits of anti-neurogenic approaches to immunotherapy (including checkpoint inhibitors already used in clinical practice).

## 6. Conclusions

Cancer cells are highly opportunistic, exploiting intrinsic developmental and defense mechanisms in their microenvironment to boost their own growth and spread. In this context, we highlighted the crucial role of peripheral nerves in the TME regulating the initiation and progression of PDAC. Importantly, sympathetic, parasympathetic, and sensory nerves each modulate distinct signaling pathways involved in tumor survival and immune escape. Therefore, exploring the potential synergistic benefits of combining nerve-targeting therapies with chemotherapy or immunotherapy, along with the development of novel anti-neurogenic approaches, could help inhibit both pancreatic cancer progression as well as cancer-related pain.

## Figures and Tables

**Figure 1 cancers-16-04260-f001:**
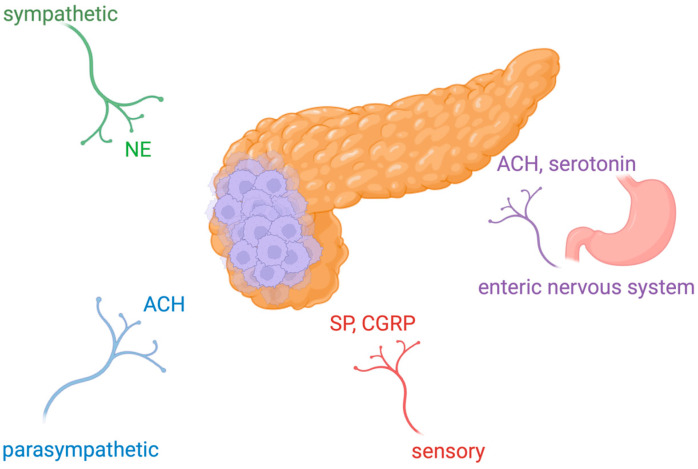
Innervation of pancreatic cancer. In pancreatic ductal adenocarcinoma, nerve fibers include axons originating from sympathetic and parasympathetic nerve fibers, sensory afferent nerve fibers, and also nerve fibers from the enteric nervous system. Neurotransmitters are released from parasympathetic fibers (ACH), sympathetic fibers (NE), sensory fibers (SP, CGRP), and enteric fibers (ACH, serotonin). Abbreviations: NE, noradrenaline; ACH, acetylcholine; SP, substance P; CGRP, calcitonin gene-related peptide. Created in BioRender. Garajova, I. (2024). https://BioRender.com/r29r619 (accessed on 11 December 2024).

**Figure 2 cancers-16-04260-f002:**
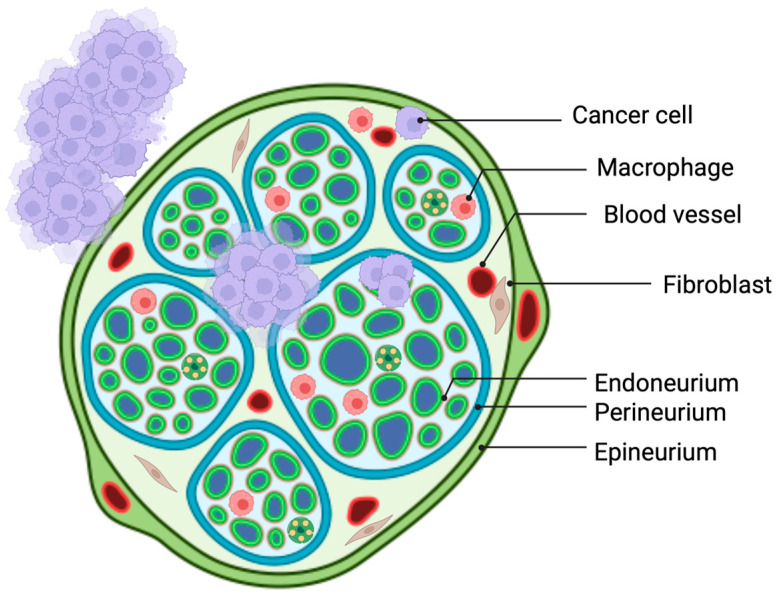
Tumor cells can be present in different spaces of the neuron sheath, including the endoneurium, perineurium, or epineurium. Created in BioRender. Garajova, I. (2024). https://BioRender.com/i31r247 (accessed on 11 December 2024).

**Figure 3 cancers-16-04260-f003:**
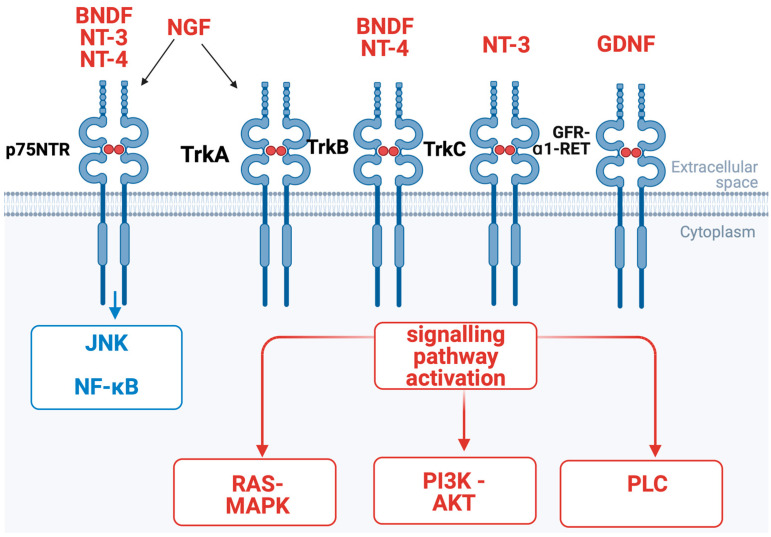
NGF/TrkA signaling pathway in PDAC. NGF acts on TrkA receptor and p75NTR receptors with subsequent activation of different signaling pathways. Abbreviations: TrkA, tropomyosin receptor kinase A; TrkB, tropomyosin receptor kinase B; TrkC, tropomyosin receptor kinase C; neurotrophin-3 (NT-3); neurotrophin-4/5 (NT-4/5); GDNF, glial-derived neurotrophic factor; NGF, nerve growth factor; MAPK, mitogen-activated protein kinase; PI3K, phosphatidylinositol-3-kinase; PLCγ, phospholipase Cγ; JNK, c-Jun N-terminal kinase; NF-κB, nuclear factor-κB; BDNF, brain-derived neurotrophic factor. Created in BioRender. Garajova, I. (2024). https://BioRender.com/s57o562 (accessed on 11 December 2024).

**Figure 4 cancers-16-04260-f004:**
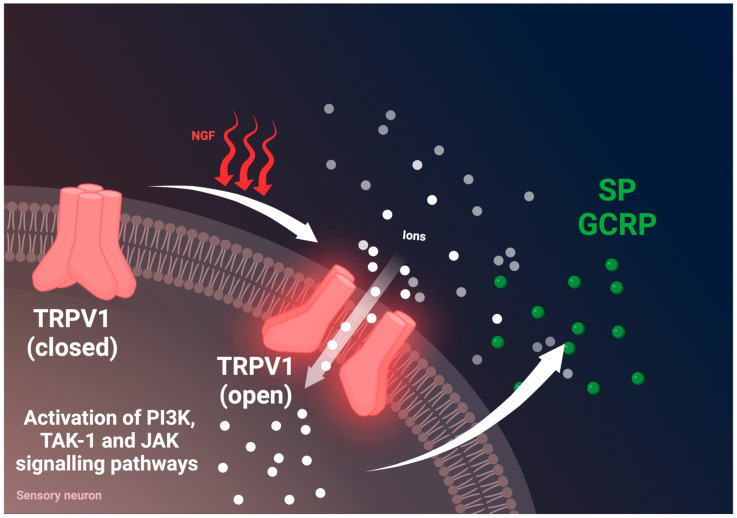
Activation of TRPV1 in sensory nerves by NGF. TRPV1 is expressed by sensory afferents that have cell bodies in vagus nerve, trigeminal ganglia, and dorsal root ganglia. TRPV1 activation initiates downstream signaling of three major pathways including PI3K, TAK-1, and JAK signaling pathways. TRPV1 activation is required for release of neuropeptides such as SP and CGRP. Abbreviations: TRPV1, transient receptor potential vanilloid; PI3K, phosphatidylinositol-3-kinase; TAK-1, transforming growth factor-activated kinase 1; JAK, Janus kinase; SP, substance P; CGRP, calcitonin gene-related peptide. Created in BioRender. Garajova, I. (2024). https://BioRender.com/z29w911 (accessed on 11 December 2024).

**Figure 5 cancers-16-04260-f005:**
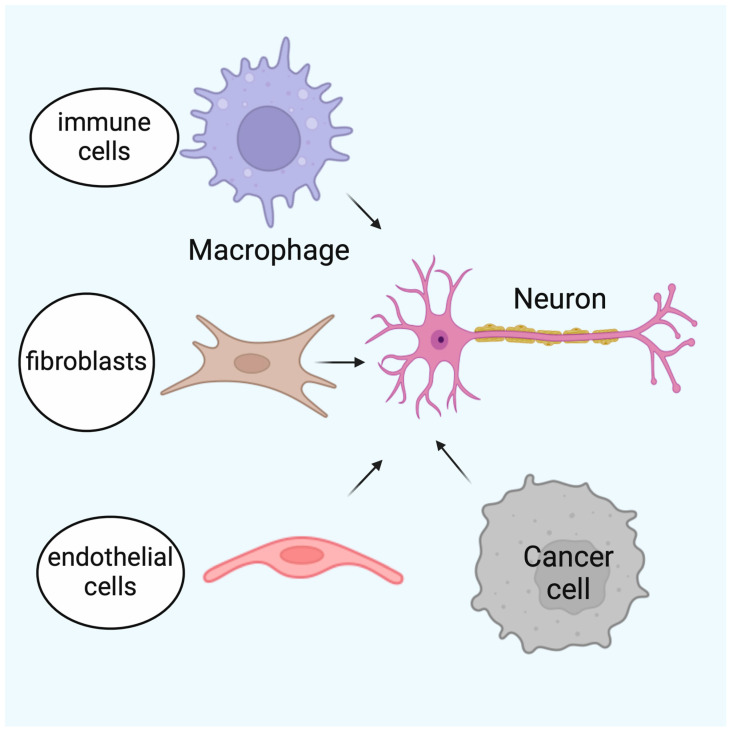
The nerves in the pancreatic cancer tumor microenvironment. The nerves influence other components of the tumor microenvironment, in particular cancer cells, immune cells, fibroblasts, and endothelial cells. Created in BioRender. Garajova, I. (2024). https://BioRender.com/k18n399 (accessed on 11 December 2024).

**Figure 6 cancers-16-04260-f006:**
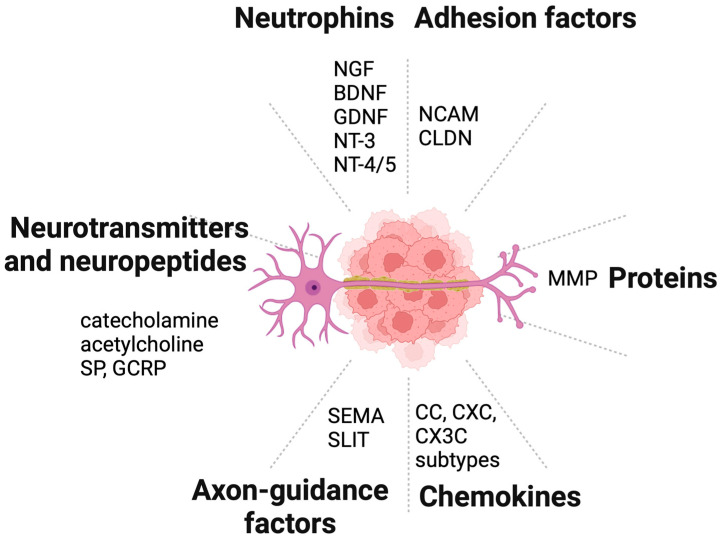
Molecules involved in neural invasion. Several molecules are involved in neural invasion in PDAC, including neutrophins, adhesion factors, proteins, chemokines, axon-guidance molecules, neurotransmitters, and neuropeptides. Created in BioRender. Garajova, I. (2024). https://BioRender.com/k54v075 (accessed on 11 December 2024).

**Figure 7 cancers-16-04260-f007:**
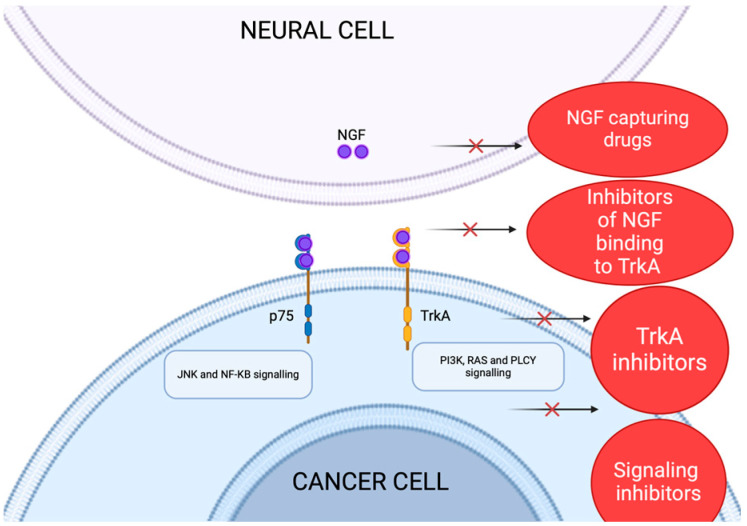
Possibilities in NGF/TrkA signaling pathway inhibition. A possible site of NGF/TrkA inhibition: (1) NGF-capturing drugs. (2) Inhibitors of NGF binding to TrkA. (3) TrkA inhibitors. (4) Inhibitors of activated signaling pathways. Created in BioRender. Garajova, I. (2024). https://BioRender.com/f74a942 (accessed on 11 December 2024).

**Table 1 cancers-16-04260-t001:** Overview of clinical trials exploring NGF-blocking antibodies for pain relief. Abbreviations: IV, intravenous; SC, subcutaneous; NSAID, non-steroidal anti-inflammatory drugs; WOMAC, Western Ontario and McMaster Universities Osteoarthritis Index.

Disease	Phase	Study Type	Intervention Arms	Main Conclusions	Refs.
Osteoarthritis	3	Multicenter, randomized, double-blind, controlled study	Tanezumab (5 mg or 10 mg) IV or placebo with or without oral naproxen (500 mg) or celecoxib (100 mg)	Tanezumab+ NSAIDs were superior over NSAIDs.	[105]
Chronic low back pain	3	Multicenter, randomized, double-blind, controlled study	Tanezumab (5 mg or 10 mg) SC or celecoxib (100 mg)	Overall adverse events (tanezumab (5 mg) = 63.0%; tanezumab 10 mg = 54.8%; celecoxib = 67.4%) and events of abnormal peripheral sensation (tanezumab 5 mg = 9.8%; tanezumab 10 mg = 4.3%; celecoxib = 4.3%). Joint safety event rates were 1.1% for tanezumab (5 mg), 2.2% for tanezumab (10 mg), and 0% for celecoxib.	
Osteoarthritis	3	Randomized, double-blind, placebo-controlled, multicenter study	Tanezumab (2.5 mg or 5 mg or 10mg) IV or placebo	Superior pain relief through tanezumab rather than through placebo.	[115]
Osteoarthritis	3	Randomized, double-blind, placebo-controlled, multicenter study	Tanezumab (5 mg or 10 mg) IV or naproxen or placebo	Better reduction in pain versus placebo.	[114]
Osteoarthritis	3	Randomized, double-blind, placebo-controlled, multicenter study	Tanezumab (5mg or 10 mg) or oxycodone (10 mg to 40 mg every 12 h) or placebo	Tanezumab demonstrated significant improvements in WOMAC pain score versus placebo and oxycodone.	[113]
Osteoarthritis	3	Randomized, double-blind, placebo-controlled, multicenter study	Diclofenac (75 mg) and tanezumab (2.5 mg or 5 mg or 10 mg) IV or placebo	Addition of tanezumab to diclofenac resulted in significant improvements in pain relief.	[112]
Diabetic peripheral neuropathic pain	2	Randomized, double-blind, placebo-controlled, multicenter study	Fulranumab (1 mg, 3 mg, or 10 mg) SC or placebo	Fulranumab (10 mg) enabled better average daily pain reduction than placebo.	[111]
Bone metastasis	2	Randomized, double-blind, placebo-controlled, multicenter study	Tanezumab (10 mg) EV or placebo	Efficient in patients with lower baseline opioid use and/or higher baseline pain.	[118]

## Data Availability

The data presented in this study are available on request from the corresponding author.

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
