# Peer review of "Targeting Perineural Invasion in Pancreatic Cancer"

_cancers, 2024, doi:10.3390/cancers16244260_

Round 1

Reviewer 1 Report

Comments and Suggestions for Authors

It was with great pleasure that I have read the excellent review by Garajova and Giovannetti entitled “Targeting perineural invasion in pancreatic cancer”.

This review is of great interest due to the truly infaust prognosis of  pancreatic ductal carcinoma (PDAC) that, even if it is not a prevalent cancer, is slated to very soon become the second leading cause cancer mortality and in a recent study was project to have the highest mortality to incidence ratio (MIR) for all cancers by 2050 (Bizuayehu, et al 2024 JAMA Network Open. 2024;7(11):e2443198. doi:10.1001/jamanetworkopen.2024.43198 ). This high mortality is due to its rapid dissemination and metastasis. As the authors state in the section 2 on page 3,  the perineural route is a newly discovered dissemination mechanism. Importantly, the perineural network also plays a large role in pain-driven cachexia, which is one of the major problems for PDAC patients.

The review is very well organized, written and informative. The Figures for the most part help in the understanding of the text and they are well organized. However, they are difficult to read because in almost every figure the writing is far too small. Just as an example, in Figure 6, there is white writing in red circles and black writing in in a beige rectangle that are practically impossible to read.

This absolutely needs to be corrected because this ruins an otherwise excellent review!!

I am wondering if a table summerizing all the factors (molecules) discussed in the text and another table summerizing all the adhesion factors that could be targeted could help in understand the complex nature of the field presented in the text. Just a suggestion and not a requirement.

In conclusion, I feel that this is a very strong review that has given important insights in the role of the perineural system in driving pancreatic cancer and potential targets within this system in on how its pharmacological inhibition could have clinical relevance.

Therefore, after the improvement of the figures suggested above, I recommend that this review be published.

Author Response

Thank you very much, we corrected the figures.

Reviewer 2 Report

Comments and Suggestions for Authors

The manuscript by Garajova and Giovannetti provides an analysis of perineural invasion (PNI) in PDAC, a phenomenon associated with tumor aggressiveness and pain. The authors review the role of peripheral nerves in the tumor microenvironment, highlight signaling pathways and molecules involved in PNI, and discuss potential therapeutic strategies.

Major points

The review explores molecular and cellular mechanisms driving PNI in PDAC, drawing on a wide range of literature. It effectively links molecular findings to clinical outcomes, emphasizing the therapeutic potential of targeting PNI. 

However, the role of NGF-TrkA signaling is revisited multiple times. To address this, consolidate discussions on neurotrophins into a single section that highlights clinical implications and therapeutic challenges.

Introduce hypotheses in targeting PNI, such as integrating immunotherapy with anti-neurogenic approaches, and discuss pros and cons. Which type of immunotherapy and why?

Summarize clinical trials involving NGF-targeting therapies in a table for better clarity.

Include images showing actual PNI, as many readers may not be familiar with it.

Page 11, line 437: The statement about NK-1R antagonists as antitumor agents requires more evidence and discussion. Instead of mentioning only its application as an antiemetic and refer to a reference, discuss its potential as a cancer therapy.

Minor points

Graphical abstract: The font size of the labels is too small to read clearly. Increase the font size for readability.

Figure 6: The labels in the red circles are difficult to read due to their small font size. Adjust the font size for clarity.

Page 11, line 439: Change "as an example, NK-1R antagonist is an aprepitant already used in oncological setting as antiemetic", change to "An NK-1R antagonist, aprepitant, is already used in oncology as an antiemetic."

Page 11, line 427: Change "Importantly, seems that darifenacin might confer resistance to chemotherapy in PDAC" to "It seems that darifenacin might confer resistance to chemotherapy in PDAC."

Recommendation:

Accept with major revisions.

Author Response

Major points

The review explores molecular and cellular mechanisms driving PNI in PDAC, drawing on a wide range of literature. It effectively links molecular findings to clinical outcomes, emphasizing the therapeutic potential of targeting PNI. 

However, the role of NGF-TrkA signaling is revisited multiple times. To address this, consolidate discussions on neurotrophins into a single section that highlights clinical implications and therapeutic challenges.

We dedicated session to this topic:  “5.1. Targeting NGF-TrkA signalling pathway and another neutrophins.”

 Introduce hypotheses in targeting PNI, such as integrating immunotherapy with anti-neurogenic approaches, and discuss pros and cons. Which type of immunotherapy and why?

We introduced this interesting hypotheses, as suggested.

Summarize clinical trials involving NGF-targeting therapies in a table for better clarity.

The Table 1 has been added.

Include images showing actual PNI, as many readers may not be familiar with it.

A new figure has been added as suggested (Figure 2).

Page 11, line 437: The statement about NK-1R antagonists as antitumor agents requires more evidence and discussion. Instead of mentioning only its application as an antiemetic and refer to a reference, discuss its potential as a cancer therapy.

We added a new paragraph for this topic in the session 5.5.

Minor points

Graphical abstract: The font size of the labels is too small to read clearly. Increase the font size for readability. Corrected.

Figure 6: The labels in the red circles are difficult to read due to their small font size. Adjust the font size for clarity. Corrected.

Page 11, line 439: Change "as an example, NK-1R antagonist is an aprepitant already used in oncological setting as antiemetic", change to "An NK-1R antagonist, aprepitant, is already used in oncology as an antiemetic." Corrected.

Page 11, line 427: Change "Importantly, seems that darifenacin might confer resistance to chemotherapy in PDAC" to "It seems that darifenacin might confer resistance to chemotherapy in PDAC." Corrected.

Reviewer 3 Report

Comments and Suggestions for Authors

From a biostats and clinical epidemiology point of view, this manuscript shows no concerns, being a well planned and realized narrative review, without any novel data. I do recognize that, just for now, it would be quite hard to infer clinical results from a preclinical/translational context like the yours. From a formal point of view, I do strongly suggest to add a full abbreviations list

Author Response

Thank you, we added a full abbreviations list as suggested.

Round 2

Reviewer 2 Report

Comments and Suggestions for Authors

Overall, the authors sufficiently responded to my comments.

However, the font size of in the graphical abstract is still too small in some parts.

There are typos (neurotransmitters and catecholamine) in the labeling of the figure “Molecules involved in neural invasion”. Please fix.

Author Response

However, the font size of in the graphical abstract is still too small in some parts.

Corrected.

There are typos (neurotransmitters and catecholamine) in the labeling of the figure “Molecules involved in neural invasion”. Please fix.

Corrected.
